

# Automatic classification of kidney CT images with relief based novel hybrid deep model

Harun Bingol[1], Muhammed Yildirim[2], Kadir Yildirim[3] and Bilal Alatas[4]

[1] Software Engineering, Malatya Turgut Ozal University, Malatya, Turkey
[2] Computer Engineering, Malatya Turgut Ozal University, Malatya, Turkey
[3] Elazig Fethi Sekin City HTRC, Elazig, Turkey
[4] Software Engineering, Firat (Euphrates) University, Elazig, Turkey

Corresponding author
Bilal Alatas, balatas@firat.edu.tr

## ABSTRACT

One of the most crucial organs in the human body is the kidney. Usually, the patient does not realize the serious problems that arise in the kidneys in the early stages of the disease. Many kidney diseases can be detected and diagnosed by specialists with the help of routine computer tomography (CT) images. Early detection of kidney diseases is extremely important for the success of the treatment of the disease and for the prevention of other serious diseases. In this study, CT images of kidneys containing stones, tumors, and cysts were classified using the proposed hybrid model. Results were also obtained using pre-trained models that had been acknowledged in the literature to evaluate the effectiveness of the suggested model. The proposed model consists of 29 layers. While classifying kidney CT images, feature maps were obtained from the convolution 6 and convolution 7 layers of the proposed model, and these feature maps were combined after optimizing with the Relief method. The wide neural network classifier then classifies the optimized feature map. While the highest accuracy value obtained in eight different pre-trained models was 87.75%, this accuracy value was 99.37% in the proposed model. In addition, different performance evaluation metrics were used to measure the performance of the model. These values show that the proposed model has reached high-performance values. Therefore, the proposed approach seems promising in order to automatically and effectively classify kidney CT images.

## INTRODUCTION

Some organs are vital in the human body. If these vital organs are missing or not functioning properly, the patient may die. Kidneys are among the first of these vital organs. Approximately 10% of adult individuals are affected by chronic kidney disease, which is among the top 20 causes of death in the world (*Al Imran, Amin & Johora, 2018*). The kidneys basically filter the blood in the body and ensure that excess water and minerals, as well as harmful substances, are excreted from the body through the urine. The consequences

of disease states that may occur in this important organ that leads to the excretory system are very serious. The main serious diseases that can be seen in the kidneys are known as chronic kidney disease (*Levey & Coresh, 2012*), acute kidney failure (*Coca, Singanamala & Parikh, 2012*), urinary tract infection (*Lee & Neild, 2007*), kidney stones (*Khan et al., 2016*), and polycystic kidney disease (*Wilson, 2004*). Depending on the type of disease, the symptoms of the patient may vary. Some of these symptoms are weakness, fatigue, confusion, edema and swelling in the body parts, skin rash and dryness, blood in the urine, pain and burning sensation during urination, muscle and joint pain, sexual problems, nausea, and vomiting.

Kidney diseases are more common in other members of the community than in some age ranges or existing diseases of people living in the community. People with diabetes or hypertension have a higher incidence of kidney disease than healthy individuals. In addition, people with kidney disease in close family members are at a higher risk of developing kidney disease genetically. Finally, elderly people are more likely to develop kidney disease than younger people.

Considering all these situations, it is extremely important for the patient to examine the kidney CT images taken from the individual and to start the treatment process by diagnosing the patient. Many patients continue their lives depending on dialysis devices as a result of losing the chance of early treatment or applying the wrong treatment methods. This creates an extremely bad situation that limits the social life of the patient.

In recent years, the diagnosis and classification of diseases with the help of computers is quite common in the medical field. Machine learning and deep learning-based technologies are at the forefront of computer-aided systems. There are many reasons for the widespread use of these technologies. One of these reasons is that the latest technologies used can help prevent the expert from making a wrong diagnosis. Another advantage is that it can diagnose patients early in rural settlements where there are not enough specialists. The accuracy rate in diagnosing chronic kidney disease by experienced nephrologists is between 60.3% and 80.1% (*Kuo et al., 2019*). The difficulty of distinguishing disease images with the naked eye and the inability of the specialist to distinguish fully or to classify even conflicting CT images with high accuracy can be shown as another important advantage of computer-aided systems.

In this study, a deep learning-based hybrid model is proposed to effectively classify kidney CT images containing cysts, stones, and tumors. Thanks to this model, kidney images are separated as training and testing. After the deep model proposed in this article is trained with training data, it is checked with test data.

The following is a summary of the study's main contributions:

●A deep learning-based hybrid model is proposed that can effectively detect and classify many kidney diseases.

●Classifying many kidney diseases separately with the proposed model will minimize the possibility of misdiagnosis and treatment.

●In addition to relieving the workload of kidney specialists, the proposed model will be able to classify even the cases that are very difficult to notice with high accuracy.

●While the highest accuracy value obtained in eight different pre-trained models was 87.75%, this accuracy value was 99.37% in the proposed model. Therefore, the proposed approach can be used to automatically and efficiently classify kidney CT images.

●Since the kidney is one of the most important organs in the human body and kidney diseases are typically noticed late by patients, these patients tend to rely on the use of a dialysis machine or, alternatively, may die. The most important contribution of this study is its classification of many kidney diseases with extremely high accuracy.

In the second section of the study, studies in the literature related to the classification of kidney diseases with artificial intelligence technologies are summarized. In the third section, the dataset used in the experiments, the classifiers, pre-trained models, and the proposed hybrid deep architecture are explained in detail. The experimental results are discussed in the fourth section. The conclusions are discussed in the fifth section. The article's findings and upcoming research are discussed in the final part.

## RELATED WORKS

In the literature, there are many studies in which artificial intelligence architectures are used in the classification of kidney diseases. Some of these studies are as follows:

*Kuo et al. (2019)* stated that they used ResNet architecture, one of the deep learning methods, to automatically classify eGFR and chronic kidney disease status using kidney ultrasound images. They used a dataset consisting of 4,505 kidney ultrasound images. They stated that the average accuracy value of the deep methods they suggested was 85.6%. Developing different models in the study can increase the performance of the paper (*Kuo et al., 2019*).

*Attia et al. (2015)* tried to classify kidney ultrasound images using artificial neural networks and principal component analysis (PCA) methods. They used the dataset consisting of five classes and 66 kidney ultrasound images in the experiments. Basically, a median filter was applied to the images for which region of interest (ROI) segmentation was performed with the help of experts. Then, the size of the obtained feature map was reduced by the PCA technique. It was stated that the obtained classification accuracy was 97% (*Attia et al., 2015*). The main shortcoming of this study is that the dataset used included very few kidney ultrasound images.

*Sudharson & Kokil (2020)* classified the well-known deep models ResNet101, MobileNetV2, and ShuffleNet architectures from the dataset consisting of 4,940 kidney ultrasound images, including four classes, by extracting features and combining them with a support vector machine. In this study, experiments were carried out using noisy images and high-quality images. The Gaussian method was used to remove noise. According to the results of the studies, the accuracy rate for classifying high-quality images was 96.54%, while the accuracy rate for classifying noisy images was 95.58%. Using optimization algorithms for feature selection in the study will make the developed model run faster (*Sudharson & Kokil, 2020*).

*Park et al. (2019)* proposed a method to generate 3-dimensional segmentation of the kidney from kidney CT images with deep learning models. It was stated that more successful results were obtained with the proposed method compared to manual segmentation.

*Selvarathi et al. (2021)* tried to classify kidney diseases with convolution neural network architectures using a dataset consisting of 50 kidney ultrasound images. They used KNN as a classifier in their study. It is emphasized that kidney ultrasound images are classified with an accuracy rate of 96.67%. The number of data used in the study is not enough for deep architectures to produce healthy results. Increasing the number of data will affect the performance of the models.

*Ma et al. (2020)* proposed a model called HMANN to detect and classify chronic kidney diseases with renal ultrasound images. They tested their proposed model on a dataset of 400 publicly available kidney images. It has been reported that the proposed HMANN method for kidney segmentation is successful in finding the exact location of the kidney stone and has an accuracy rate of 97.5% in the SVM classifier. The biggest limitation of the study is the limited number of data. It is thought that increasing the number of data will affect the performance of the models.

*Abdeltawab et al. (2021)* proposed a deep model that can distinguish between tumor-causing tumors and tumor-free areas using kidney ultrasound images. They used 64 full-slide images during the experiments. Additionally, they claimed that the deep model they put forth could effectively differentiate between clear cell papillary renal cell carcinoma and clear cell renal cell carcinoma. In addition, the map consistency was achieved by applying the Gauss-Markov random field smoothing method to the kidney images. They stated that the classification accuracy value of the method they suggested was 92% (*Abdeltawab et al., 2021*).

*Yildirim et al. (2021)* classified whether there were stones in 1,799 kidney CT images by using deep learning techniques. The deep model they proposed indicated that even stone images with small dimensions can be classified effectively. It was stated that the accuracy rate in classifying kidney CT images was 96.82%.

*Baygin et al. (2022)* used a two-class dataset consisting of 1,799 kidney CT images. In the study, it was aimed to detect stone from a two-class data set. In the study, a model named ExDark19 was developed. In the developed model, DarkNet19 architecture was used as the basis for feature extraction, and feature selection was made with the INCA method. For the classification process, the KNN classifier was preferred. Researchers have achieved an accuracy value of over 99% in this study they have done for the classification of kidney stones.

*Caglayan et al. (2022)* explained their study's purpose as using deep learning models to detect kidney stones in different planes according to stone size using CT images. In the study, CT images of 455 patients were used. Researchers have reached an accuracy rate of 99.1% in this study for detecting kidney stones.

In the study performed by *Islam et al. (2022)*, vision transformers and pre-trained architectures were used to classify the images in the dataset consisting of 12,446 images and four classes. ResNet, VGG16, and Inception v3 are pre-trained models used for the classification of CT images. The researchers stated that the most successful result was obtained in the Swin transformer model (99.1% accuracy) in their study and that the training of this model was shorter than the other models used.

## THEORETICAL BACKGROUND

### Dataset

The dataset used during the experiments includes kidney CT images consisting of four classes: cyst, normal, tumor, and stone. The dataset was collected from different hospitals in Bangladesh. Both contrast and non-contrast techniques were used to create of the dataset. The data of each patient was obtained using Dicom images and these Dicom images were then converted to .jpg image format. After this process, the images were under the supervision of a radiologist to confirm their accuracy (*Islam et al., 2022*). In the dataset used in the study, there are 12,446 images, a total of four classes. 3,709 of these images belong to the cyst, 2,283 tumor, 1,377 stone, and 5,077 normal classes.

### Pre-trained deep models and machine learning classifier

In the study, eight pre-trained deep models were used to compare the performance of the developed model in the classification of kidney CT images. The AlexNet architecture was proposed by *Krizhevsky, Sutskever & Hinton (2012)*. This architecture was selected as the winner of the ImageNet competition held in 2012. AlexNet accepts 227x227 images as input. Also, this architecture has 61 million parameters (*Krizhevsky, Sutskever & Hinton, 2012*). Densenet201 architecture was proposed by *Huang et al. (2017)*. In this architecture, each layer has a working logic that accepts the feature maps of all previous layers as input. The Efficientnet architecture was proposed by *Tan & Le (2019)*. The main feature of this architecture is the balanced scaling of both the depth and width of the mesh and the resolution dimensions by using an effective composite coefficient (*Tan & Le, 2019*). The GoogleNet deep model was proposed by *Szegedy et al. (2015)*. Deep model won the ImageNet competition with a 6.66% error rate. The main difference of this deep model is that a modular structure is used in parallel to decrease the memory cost of the neural network and to reduce the probability of over-learning of the neural network (*Szegedy et al., 2015*). The Inception architecture was proposed by *Szegedy et al. (2016)*. In this deep architecture, it consists of three basic parts, starting block, convolution block, and classifier. The MobileNet model was developed by *Howard et al. (2017)*. This model is mostly built to be used in display applications using portable and embedded systems, in other words, to get effective results on devices with low hardware. ResNet50 was proposed by *He et al. (2016)*. The main feature of this architecture is that it uses residual blocks to figure out the problem of loss of gradients in the neural network. As a result of this process, the gradient problem, which decreases continuously and approaches zero, is solved. The ShuffleNet was developed by *Zhang et al. (2018)*. The main feature of this model is that effective results can be obtained on portable devices with limited information processing capacity such as the MobileNet architecture.

The feature map obtained with the hybrid model that is proposed in this study was given as input to the wide neural network (*Lee et al., 2019*) classifier.

### Proposed CNN-based hybrid model

In the study, a CNN-based deep model was developed to classify kidney CT images. In the proposed model, seven convolution layers, four batch normalization layers, seven

Relu activation functions, seven maxpooling, one fully connected, one softmax and one classification layer are used. The size of the images coming to the model is automatically converted to $227 \times 227 \times 3$ format in the input layer. In the proposed model, features obtained from the 6th and 7th convolution layers are combined. Then, the obtained feature map was optimized with the relief dimension reduction method so that the model could produce more successful results in a faster time. In the last step, the optimized feature map is classified in the wide neural network. The flow chart of the proposed model is given in Fig. 1.

In the proposed model, kidney CT images are given to the proposed model *via* the input layer. Then, the convolution layer was used to extract feature maps from the CT images in the dataset. In this layer, filters are moved on the image and it is tried to determine whether a certain attribute is in the input. At the end of these processes, the input is passed through certain filters and a new appearance, namely its properties, emerges. These features are called feature maps. In addition, with the filters applied to the input image, the depth of the network is increased and it is aimed that the network will give more accurate results. Each filter in the convolution layers is applied to the image obtained from the previous convolution layer, reducing the aspect ratio of the input image and increasing its depth. Another layer used in the proposed model is the pooling layer. One of the purposes of pooling is to reduce the size of feature maps and reduce diversity. In this way, the computational costs and input sizes of the next layer are reduced, while retaining important features. It also prevents network memorization of the jointing process. The network layers do not need to wait for the preceding tier to catch up because of the batch normalization layer. It enables concurrent learning. As a result, training might go more quickly. In the suggested paradigm, Relu serves as the activation function. In CNN architectures, the Relu activation function is frequently employed. Thanks to the Relu function, it is ensured that the function takes the value 0 on the negative axis. The goal here is to make the network run faster. The disadvantage of this is that since the derivative is zero in this region, learning is also prevented.

Feature maps were obtained by using the 6th and 7th convolution layers of the proposed CNN-based model in the study. The size of the feature map obtained from the convolution 6 layer is $12,446 \times 384$, while the size of the feature map obtained from the convolution 7 layer is $12,446 \times 256$. Among the feature maps obtained from both convolution layers, the best 200 features were selected using the relief method. The obtained features were combined and a feature map of $12,446 \times 400$ size was obtained. Finally, the optimized and combined feature map is classified using a wide neural network. In summary, the proposed model first extracted feature maps over 2 different layers. Extracted feature maps were combined after optimizing. The wide neural network then classifies the optimized feature map. The layers, activation functions, and parameter values used in the proposed model are presented in Table 1.

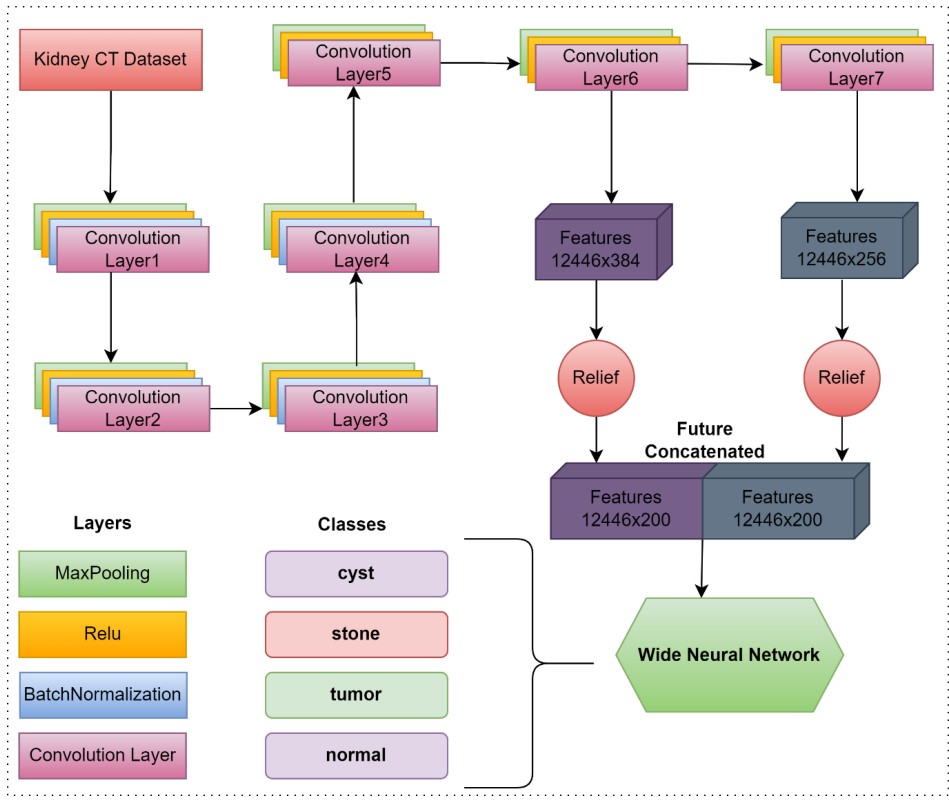

**Figure 1** The proposed hybrid model's flowchart.

## RESULTS

During the experiments, fixed parameters were used for all pre-trained deep models. Again, all these experimental processes were performed on the same computer. A computer with Windows 10 operating system, MATLAB 2021b program, Intel i5 processor, 16GB RAM, 4GB graphics card, 500GB SSD disk were used in the experiments. Some metrics were considered to compare the performance of deep models. These metrics are as follows: Accuracy (Acc), Specifty (Spc), Sensitivity (Sens), Precision (Pre), F-score (F1), False Discovery Rate (FDR), False Negative Rate (FNR), and False Positive Rate (FPR) (*Bingol, 2022*).

The classes Cyst, Normal, Stone, and Tumor were given in the confusion matrix by the numbers 1, 2, 3, and 4, respectively.

### Pre-trained CNN models

Examining the literature shows that there are numerous deep models. However, eight of the most well-known deep models were used during the experiments. In order to compare deep architectures with each other, it is important to run them in the same environment and with the same parameters. Application results in the MATLAB environment, MaxEpochs value is 5, LearningRate 1e−4, and MiniBatchSize value is 16.

**Table 1  Characteristics of the proposed hybrid model.**

|  | Name | Type | Activations | Learnables |
|---|---|---|---|---|
| 1 | imageinput | Image Input | 227x227x3 | – |
| 2 | conv_1 | Convolution | 55x55x96 | Weights 11x11x3x96 Bias 1x1x96 |
| 3 | batchnorm_1 | Batch Normalization | 55x55x96 | Offset 1x1x96 Scale 1x1x96 |
| 4 | relu1 | ReLU | 55x55x96 | – |
| 5 | maxpool_1 | Max Pooling | 27x27x96 | – |
| 6 | conv_2 | Convolution | 27x27x256 | Weights 3x3x96x256 Bias 1x1x256 |
| 7 | batchnorm_2 | Batch Normalization | 27x27x256 | Offset 1x1x256 Scale 1x1x256 |
| 8 | relu2 | ReLU | 27x27x256 | – |
| 9 | maxpool_2 | Max Pooling | 13x13x256 | – |
| 10 | conv_3 | Convolution | 13x13x384 | Weights 3x3x256x384 Bias 1x1x384 |
| 11 | batchnorm_3 | Batch Normalization | 13x13x384 | Offset 1x1x384 Scale 1x1x384 |
| 12 | relu3 | ReLU | 13x13x384 | – |
| 13 | maxpool_3 | Max Pooling | 6x6x384 | – |
| 14 | conv_4 | Convolution | 6x6x512 | Weights 3x3x384x512 Bias 1x1x512 |
| 15 | batchnorm_4 | Batch Normalization | 6x6x512 | Offset 1x1x512 Scale 1x1x512 |
| 16 | relu4 | ReLU | 6x6x512 | – |
| 17 | maxpool_4 | Max Pooling | 3x3x512 | – |
| 18 | conv_5 | Convolution | 2x2x512 | Weights 3x3x512x512 Bias 1x1x512 |
| 19 | relu5 | ReLU | 2x2x512 | – |
| 20 | maxpool_5 | Max Pooling | 1x1x512 | – |
| 21 | conv_6 | Convolution | 1x1x384 | Weights 3x3x512x384 Bias 1x1x384 |
| 22 | relu6 | ReLU | 1x1x384 | – |
| 23 | maxpool_6 | Max Pooling | 1x1x384 | – |
| 24 | conv_7 | Convolution | 1x1x256 | Weights 3x3x384x256 Bias 1x1x256 |
| 25 | relu7 | ReLU | 1x1x256 | – |
| 26 | maxpool_7 | Max Pooling | 1x1x256 | – |
| 27 | fc_1 | Fully Connected | 1x1x4 | Weights 4x256 Bias 4x1 |
| 28 | prob | Softmax | 1x1x4 | – |
| 29 | classoutput | Classification Output | 1x1x4 | – |

**Table 2  Accuracy results of CNN models.**

| EfficientNetb0 | MobileNetv2 | ShuffleNet | AlexNet | ResNet50 | DenseNet201 | GoogleNet | Inceptionv3 |
|---|---|---|---|---|---|---|---|
| 81.52% | 82.20% | 78.99% | 75.25% | 86.06% | 84.45% | 80.72% | **87.75%** |

CNN architectures convert the size of incoming images into a common input format in the input layer. 20% of the dataset containing kidney CT images is reserved for testing and 80% is reserved for training the model. The accuracy values obtained from the pre-trained CNN models are given in Table 2.

After fine-tuning the Inceptionv3 architecture, 87.75% accuracy was achieved. This accuracy value is the highest obtained among the pre-trained deep models. The AlexNet architecture yielded the lowest accuracy value of 75.25%. Confusion matrices of all these pre-trained deep models are demonstrated in Table 3.

**Table 3  Confusion matrices obtained from deep models.**

| AlexNet | | | | | MobileNetv2 | | | |
|---|---|---|---|---|---|---|---|---|
| **1** | 630 | | 107 | 5 | **1** | 627 | | 112 | 3 |
| **2** | 2 | 824 | 115 | 74 | **2** | | 921 | 71 | 23 |
| **3** | 71 | 12 | 192 | | **3** | 54 | 14 | 207 | |
| **4** | | 230 | | 227 | **4** | | 166 | | 291 |
| | **1** | **2** | **3** | **4** | | **1** | **2** | **3** | **4** |

| ResNet50 | | | | | DenseNet201 | | | |
|---|---|---|---|---|---|---|---|---|
| **1** | 693 | | 45 | 4 | **1** | 686 | | 56 | |
| **2** | | 1,003 | 7 | 5 | **2** | | 973 | 18 | 24 |
| **3** | 63 | 3 | 208 | 1 | **3** | 109 | 6 | 155 | 5 |
| **4** | | 219 | | 238 | **4** | | 169 | | 288 |
| | **1** | **2** | **3** | **4** | | **1** | **2** | **3** | **4** |

| ShuffleNet | | | | | GoogleNet | | | |
|---|---|---|---|---|---|---|---|---|
| **1** | 659 | | 76 | 7 | **1** | 721 | | 21 | |
| **2** | 2 | 888 | 64 | 61 | **2** | | 1,006 | 8 | 1 |
| **3** | 61 | 28 | 186 | | **3** | 94 | 9 | 172 | |
| **4** | | 215 | 9 | 233 | **4** | 2 | 345 | | 110 |
| | **1** | **2** | **3** | **4** | | **1** | **2** | **3** | **4** |

| Inceptionv3 | | | | | EfficientNetb0 | | | |
|---|---|---|---|---|---|---|---|---|
| **1** | 671 | | 71 | | **1** | 676 | 3 | 63 | |
| **2** | | 986 | 14 | 15 | **2** | | 929 | 74 | 12 |
| **3** | 55 | 3 | 216 | 1 | **3** | 78 | 13 | 183 | 1 |
| **4** | | 146 | | 311 | **4** | | 216 | | 241 |
| | **1** | **2** | **3** | **4** | | **1** | **2** | **3** | **4** |

When the complexity matrix shown in Table 3 of the Inceptionv3 architecture, which was more successful than the other seven deep models with an accuracy value of 87.75%, was examined, 2,184 of the 2,489 test images were classified correctly and 305 were classified incorrectly.

Inceptionv3 architecture correctly classified 671 of 742 cyst images and classified 71 as stone. While it correctly classified 986 of 1,015 normal images without any kidney disease, it misclassified 14 as stone and 15 as tumor. While it classified 216 of 275 stone images correctly, it classified 55 as cyst, three as normal and one as tumor incorrectly. Again, while this model correctly classified 311 of 457 tumor images, it misclassified 146 as normal, that is, disease-free image class. In addition, when the complexity matrix of Alexnet, which achieved the lowest accuracy value with 75.25%, was examined, 1,873 of the 2,489 test images were classified as correct and 616 of them were incorrectly classified. Deep models, by their nature, do not show the same success on every dataset. In order to demonstrate the effectiveness of the suggested approach, experiments were conducted using eight distinct deep models.

## Proposed model

It has been proposed to identify kidney CT images using a hybrid model based on CNN. The proposed model's performance was compared to those of pre-trained models widely used in related studies and the literature. Table 4 contains the confusion matrices that were obtained using the proposed model.

Table 4 shows that 12,368 of the 12,446 kidney images were correctly classified by the proposed model, whereas 78 of them were wrongly classified. While the proposed model predicted 3,692 of 3,709 cyst images correctly, it predicted 17 of them incorrectly. Similarly, the proposed model correctly classified 5,065 of 5,077 images labeled normal, while it classified seven incorrectly. There are 1,377 images in the stone class. While the proposed model predicted 1,348 of these images correctly, it predicted 29 of them incorrectly. Finally, 2,263 of the 2,283 images in the tumor class were predicted correctly by the proposed model, while 20 of them were predicted incorrectly. Comparison of the proposed model with the accepted pre-trained models in the literature is given in Table 5.

When Table 5 is analyzed, it can be noted that the InceptionV3 architecture achieved the best accuracy value among the pre-trained models with 87.75%. The proposed model performs 99.37% in the same dataset. When it came to categorizing kidney CT images, the suggested model performed better than the pre-trained models accepted in the literature. Table 6 provides performance evaluation metrics for the suggested model.

The class in which the proposed model is most successful is the Normal class with an accuracy rate of 99.76%, while the class in which it fails the most is the Stone class with an accuracy rate of 97.89%. The average accuracy of the developed model is 99.37%. The AUC curves get in the proposed model are given in Fig. 2.

## DISCUSSION

Kidney CT images are generally used in the diagnosis of kidney diseases (*Chen et al., 2020*). Urine analysis is also often requested by specialists to measure whether the kidneys are working properly (*Levey & James, 2017*). The most common diseases in the kidneys are cysts, stones and tumors. These can be very painful and can lead to the life of the patient in terms of their consequences. Although kidney imaging devices have developed with the development of technology, the interpretation of these images by experts is extremely important. In recent years, diagnosis of diseases with artificial intelligence based on deep learning technologies has been realized in many medical fields (*Bingol & Alatas, 2021*; *Yildirim et al., 2022*; *Diker et al., 2021*).

In this study, a new model based on deep learning has been proposed for the diagnosis and classification of kidney diseases. In the proposed deep model, seven convolution layers, four batch normalization layers, seven Relu activation functions and seven maxpooling are used. In addition, the features obtained from the 6th and 7th convolution layers of the model were combined. The feature map produced in this way was then subjected to the relief approach. Then, the reduced feature map was classified in classifiers. One of the biggest features of the proposed model is the low number of layers. The success rate of the proposed model is 99.37%, with higher accuracy than the best-known deep learning

**Table 4  The confusion matrix obtained from the proposed model.**

| Proposed model | | | | |
|---|---|---|---|---|
| 1 | 3,692 | | 13 | 4 |
| 2 | 1 | 5,065 | 7 | 4 |
| 3 | 18 | 8 | 1,348 | 3 |
| 4 | 10 | 10 | | 2,263 |
| | 1 | 2 | 3 | 4 |

**Table 5  Comparison of the proposed model with pre-trained models.**

| EfficientNetb0 | MobileNetv2 | ShuffleNet | AlexNet | ResNet50 | DenseNet201 | GoogleNet | Inceptionv3 | Proposed Model |
|---|---|---|---|---|---|---|---|---|
| 81.52% | 82.20% | 78.99% | 75.25% | 86.06% | 84.45% | 80.72% | **87.75%** | **99.37**% |

**Table 6  Performance evaluation metrics of the proposed model.**

| Classes | Acc. | Spc. | Sens. | F1 | FDR | FNR | FPR |
|---|---|---|---|---|---|---|---|
| Cyst | 99.54 | 99.80 | 99.22 | 99.38 | 0.45 | 0.77 | 0.19 |
| Normal | 99.76 | 99.83 | 99.64 | 99.70 | 0.23 | 0.35 | 0.16 |
| Stone | 97.89 | 99.73 | 98.53 | 98.21 | 2.10 | 1.46 | 0.26 |
| Tumor | 99.12 | 99.80 | 99.51 | 99.31 | 0.87 | 0.48 | 0.19 |

architectures, and has reached highly competitive results. Thanks to this model, if there is a possible kidney disease even in patients who have routine check-ups, the disease will be detected early and treatment will be started early. Studies on the classification of kidney diseases are listed in Table 7.

The biggest limitation of this study is that the proposed model does not work online over the Internet. Among the studies planned for future studies, kidney CT images will be tried to be obtained from hospitals in different parts of the world. In this way, both a larger dataset will be created and the limitation that is mentioned in this article will be avoided. In addition, in the future studies, using meta-heuristic methods will increase the performance of this model by obtaining more useful features in the optimization phase.

## CONCLUSIONS

The kidney in the human body is a vital organ that is directly affected by nutrition and natural conditions. It is a fact accepted by the scientific world that more kidney diseases are seen in regions where access to clean drinking water is limited. The most important and difficult common aspects of kidney diseases are that they are noticed by the patient and treatment is started after the disease has progressed. In such cases, the treatment is either unsuccessful or the treatment process takes a long time. Thus, both the patient suffers and the treatment costs increased considerably. Thanks to computer-aided systems, it is possible to reduce both patient grievances and treatment costs. Therefore, artificial intelligence-based model developed in this study, has been achieved an accuracy value

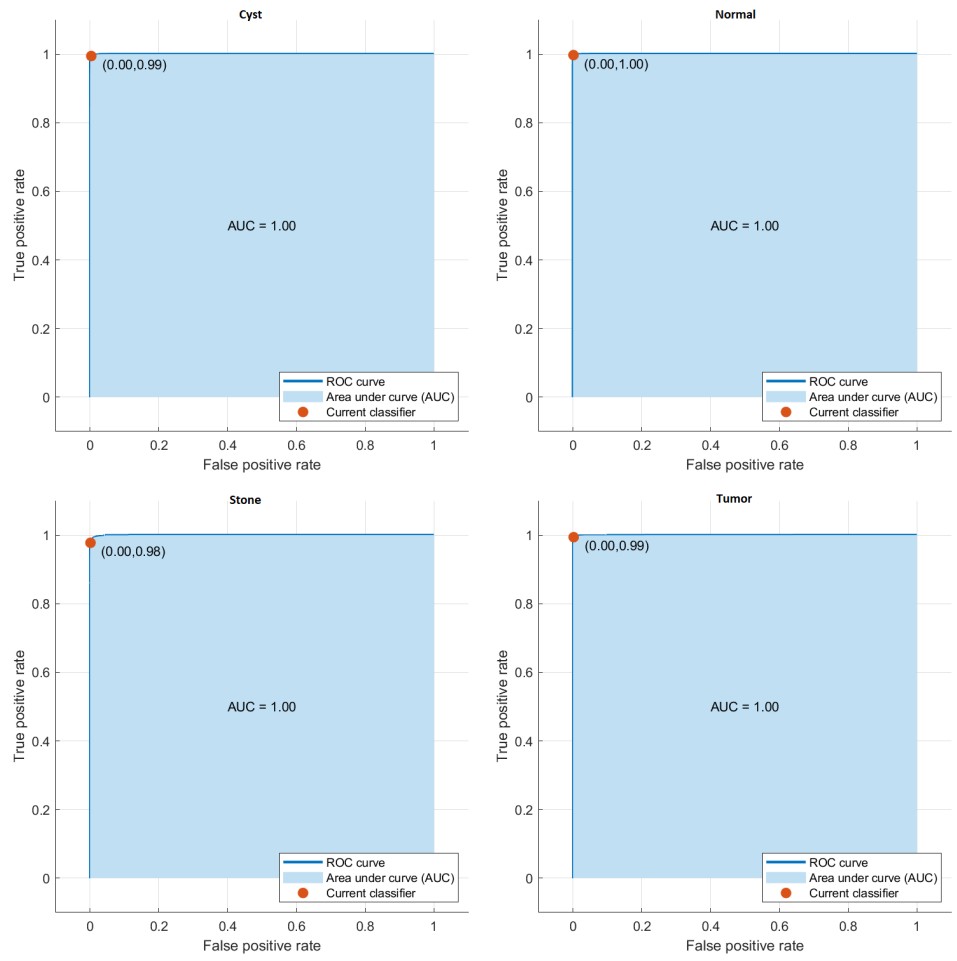

**Figure 2** AUC curves of the proposed model.

**Table 7** Studies on classification of kidney images.

| Reference | Method | Number of Images | Acc.(%) |
|---|---|---|---|
| *Kuo et al. (2019)* | Resnet | 4,505 | 85.60 |
| *Attia et al. (2015)* | ROI +Median Filter+PCA | 66 | 97.00 |
| *Sudharson & Kokil (2020)* | Resnet101+MobilenetV2+Shufflenet+SVM | 4,940 | 96.54 |
| *Selvarathi et al. (2021)* | CNN+KNN | 50 | 96.67 |
| *Ma et al. (2020)* | HMANN+SVM | 400 | 97.50 |
| *Abdeltawab et al. (2021)* | Gauss-Markov+Resnet18 | 64 | 92 |
| *Yildirim et al. (2021)* | Grad-CAM, CNN | 1,799 | 96.82 |
| *Baygin et al. (2022)* | Darknet19+INCA,KNN | 1,799 | 99 |
| *Caglayan et al. (2022)* | Deep Learning Models | 455 | 99.1 |
| *Islam et al. (2022)* | Swin transformer | 12,446 | 99.3 |
| Proposed Model | Proposed CNN model+Relief+SVM | 12,446 | 99.37 |

of 99.37%. This result demonstrates the utility of the kidney disease classification and diagnosis method proposed in this study.

## ACKNOWLEDGEMENTS

The authors thank the owners of the dataset for sharing their data.

### Funding
The authors received no funding for this work.

### Competing Interests
Bilal Alatas is an Academic Editor for PeerJ.

### Author Contributions
- Harun Bingol conceived and designed the experiments, performed the experiments, analyzed the data, prepared figures and/or tables, and approved the final draft.
- Muhammed Yildirim conceived and designed the experiments, performed the experiments, analyzed the data, performed the computation work, authored or reviewed drafts of the article, and approved the final draft.
- Kadir Yildirim conceived and designed the experiments, analyzed the data, authored or reviewed drafts of the article, and approved the final draft.
- Bilal Alatas performed the experiments, analyzed the data, prepared figures and/or tables, authored or reviewed drafts of the article, and approved the final draft.

### Data Availability
The data is available at Kaggle: https://www.kaggle.com/datasets/nazmul0087/ct-kidney-dataset-normal-cyst-tumor-and-stone.

### Supplemental Information
Supplemental information for this article can be found online at http://dx.doi.org/10.7717/peerj-cs.1717#supplemental-information.

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
