# Peer review of "Automatic classification of kidney CT images with relief based novel hybrid deep model"

_PeerJ Computer Science, doi:10.7717/peerj-cs.1717_

## Round 0.1 · original submission · Major Revisions

- The main outstanding issue with this paper is about the contribution to the field. Please discuss the research gap well and how the proposed model contributed to solve this/these issues.
- Describe the proposed hybrid model well. Explain the used datasets (CT Scan images or/and ultra sound images).
- Evaluation should include a comparison with the state of methods.

Please address all the reviewers' comments.

Reviewer 1 ·

Basic reporting

The introduction needs to prove the limitations of the existing solutions.
The article needs to be checked by Engish proofreading. ( as in line 233 and others "that are sent to them as". avoid using "them"). and many parts of the article.
Avoid using pronouns.
Avoid using the word "proposed" for the existing solutions in the related works.
Related works miss the critical analysis.

Experimental design

The proposed model presented is not clear. Add more details on how the model is working.
What are the inputs (the structure of the inputs)?
Suggest adding an example to show the process of the model.
Why does the proposed model use 6-7 layers? not less/not more?

Validity of the findings

Why is the number of images used by each model in Table 10 not the same? not fare to compare the accuracy between the models with different images type and numbers.
What are the input dimensions for the proposed model in Table 4 and Table 5?
Discuss the findings and why got that accuracy.

Reviewer 2 ·

Basic reporting

1- The manuscript needs to be proofread by a fluent English speaker or any editing facility for the purpose of improving the language spoken.

2- Literature is missing many of the state-of-the-art studies that are directly related to the current study.

3-Figures need to be larger for better resolutions and some other Figures are not clear enough.

4-Some tables are not necessary at all.

5-Some experimental settings and techniques need to be discussed in a broader way, especially with an accuracy ~100%!

Experimental design

1-The research gap is not well-identified as the limitations of the previous trials were not mentioned from different aspects that this study might help filling the research gap.

2-Contributions are leaking novelty and they need to be revised are they seem to be a listing of the organization of the paper than highlighting the novel contributions of the study.

Validity of the findings

How do the authors ensure that they are not overfitting the data or they are facing a data leakage? Results are very high and near a 100%, which is dangerous and questionable!! It might be a breakthrough and might be overfitting. More proof is still needed through trying different methods of cross-validation, test on a new unseen bench mark data, using drop-out fraction technique, etc.

Additional comments

Abstract:
1- The authors mentioned "The proposed hyprid model" without stressing on the novelty or the contributions of that model. Elaborating on the structure is not the key point.

2- More about the results have to be mentioned using other well-known metrics e.g., sens, spec, F1-score, AUC, etc.

Introduction:

1-"help preventing" instead of "prevent"
2- Please be consistent "computer-aided" or "computer aided"
3-Contributions are leaking novelty and they need to be revised are they seem to be a listing of the organization of the paper than highlighting the novel contributions of the study.

Related work:

1-The literature is weak and needs to be updated. It is missing some of the state-of-the-art techniques that has been recently performed in the area of CKD, acute rejection, and renal cancer diagnosis using AI. Please revise

2-The authors have to motivate the work they are proposing by drawing the line that initiate the idea and encourage them to investigate other than the importance of the medical problem. In other words, what are the limitations of the previous studies and they can solve by their proposal? Is it just accuracy? Speed of diagnosis? Automation? Computational cost and complexity? The authors have to elaborate more about these issue to attract the reader.

Theoretical Background

(Dataset)
1- More details about the dataset are required.


(Methods)

1-How the authors ensure that there is no overfitting or dataleakage need to be discussed.

Figures and Tables:

1- Figure 1: Please contour or use arrows to specify what you need to show here? Also, images need to be larger for better resolution.

2- Figures 2 and 3: Please provide higher resolution for clarity.

3- Table 3: Do the authors really need such a table or just mentioning such parameters in the context will be enough?

4- Table 5: I think this table is not needed as this information is provided later on in Table 7.

Annotated reviews are not available for download in order to protect the identity of reviewers who chose to remain anonymous.

Reviewer 3 ·

Basic reporting

AI is playing a great role in early detection of several different diseases. The paper is a valuable addition to this line of enquiry. The paper is well researched and the authors presented a novel approach to present a model for the detection of the kidney disease.

Experimental design

The authors used a CNN based hybrid model on a dataset of CT scans and compared its performance with some of the existing pre-trained models.

The authors outlined their proposed hybrid model and used the existing dataset with same parameters are to test the models presented in the existing studies and compared with their proposed model.

It has been claimed that the dataset used to conduct these experiment contains CT Scan images while in the Discussion section the authors started discussing ultra sound images. This dichotomy needed to clarified.

Validity of the findings

The findings of the research are well presented but it would have been better if these findings would have been compared and contratsed with the findings of the existing literatre.

Additional comments

The paper is well presented but occasional linguistic errors are very annoying. The paper need to be thoroughly proof read by a proficient English speaker.

---

## Round 0.2 · Major Revisions

Dear authors,

Thank you for submitting the revised version of the paper. However, the response for many comments were too general. It was not clear where you added these corrections. You should include section/page you revised for each comment.

For now, please provide detailed response for these comments:

- As the authors used pre-trained CNN. Please highlight the new contribution of this paper (could be by: Explaining the hybrid nature of the proposed model)
- The 2nd and 3rd contributions mentioned in Introduction are the outcomes of applying the 1st one. Please update this part by addition the proposed method(s) not its implications.

- The response for the below comment was not clear. Indicate where you discussed the previous studies to highlight the research gap (up t0 date references should be added):

The research gap is not well-identified as the limitations of the previous trials were not mentioned from different aspects that this study might help filling the research gap.

- More details are needed to ensure that they are not overfitting the data.

- Again, this comment should be addressed well :

"The authors mentioned "The proposed hyprid model" without stressing on the novelty or the contributions of that model. Elaborating on the structure is not the key point".

- Also, these comments need to be addressed well (indicate where you update the paper):

1-The literature is weak and needs to be updated. It is missing some of the state-of-the-art techniques that has been recently performed in the area of CKD, acute rejection, and renal cancer diagnosis using AI. Please revise

2-The authors have to motivate the work they are proposing by drawing the line that initiate the idea and encourage them to investigate other than the importance of the medical problem. In other words, what are the limitations of the previous studies and they can solve by their proposal? Is it just accuracy? Speed of diagnosis? Automation? Computational cost and complexity? The authors have to elaborate more about these issue to attract the reader.

- This comment need to be addressed in more details:

It has been claimed that the dataset used to conduct these experiment contains CT Scan images while in the Discussion section the authors started discussing ultra sound images. This dichotomy needed to clarified.

- A comparison with the state of the art methods need to be added and discussed well to show the enhancements obtained by the proposed model.

---

## Round 0.3 · Major Revisions

Dear authors,

Thank you for providing the revised version of your paper. The reviewers still have a few concerns about the quality of the revision done. Please consider these comments and pay more attention to:
- highlighting the paper contributions of this paper. You may need to explain the hybrid nature of the proposed model.
- add more up-to-date references and discuss the research problems we'll.
- improve the quality of all figures used.
- proofread the paper to improve the English writing. Editing the paper with the help of a professional proofreader will be useful.

Kind regards,
Faisal Saeed
Academic Editor

**Language Note:** The Academic Editor has identified that the English language must be improved. PeerJ can provide language editing services - please contact us at [email protected] for pricing (be sure to provide your manuscript number and title). Alternatively, you should make your own arrangements to improve the language quality and provide details in your response letter. – PeerJ Staff

Reviewer 1 ·

Basic reporting

The revised version is clear and unambiguous, and professional English used throughout

Experimental design

Acceptable

Validity of the findings

All underlying data have been provided; they are robust, statistically sound, & controlled.

Additional comments

The authors enhance and revised the manuscript based on the given comments

Reviewer 2 ·

Basic reporting

My comments are combined below

Experimental design

My comments are combined below

Validity of the findings

My comments are combined below

Additional comments

I still have the same concerns of:
The authors did not do their best towards enhancing the quality of the manuscript. Even the point-wise response is not handeled well in an organized and clear manner. Their responses are not elaborated for convincing the reviewer, instead they were conservative in their responses without much details. They did not even highlighted the changes in the PDF, instead they used the track changes in the word file, which is hard to follow. They should have put more effort in polishing the final output to convinve the reviewer about the work that have been done. I have to reject the manuscript for the following specific reasons:
1-Weak language spoken
2-Missing strong literature review and research motivation
3-The quality of the Figures are still poor, even Figure 1 when asked about adding arrows, the authors decided to remove it. It looks like the medical background is not sufficient enough!!
4-The authors also insist in keeping so many unneeded Tables that could be either merged or removed to avoid unnecessary repetition.
5-Results are doubtable and the authors did not even try to do an additional experiment of cross-validation other than the already existed 5-fold one.

Reviewer 3 ·

Basic reporting

The kidney is one of the vital organs of the human body and recent changes in the environment and lifestyle gave rise to several diseases which directly affect the functioning of the kidneys. Kidneys are affected by the presence of stones, cysts, or tumors. Kidneys are the soft tissue structures in the human body and clinicians around the world use Computer Tomography images to detect the health of the kidneys and the presence of disease-causing structures. The authors developed a CNN-based hybrid model to classify CT scan mages of the kidneys.

The literature review focused on the usage of machine learning, and deep learning for the early detection of kidney disease. In my opinion, the authors should try to expand by exploring the usage in other medical images like brain scans, etc.

Experimental design

The authors used Krizhevsky's proposed Alexnet architecture. This is an innovative approach and the authors employed this technique eloquently.

The authors need to expand the discussion.
In lines 337-339, the authors mentioned the limitations which are contradicting the facts mentioned in lines 145-150. In lines 145-150 authors said that the images were taken from different hospitals and in lines 337-339 mentioned that the lack of regional variation is a limitation.
The discussion and conclusion need to be redrafted and authors need to highlight the achievements.

Validity of the findings

The authors need to redraft the discussion and conclusion to explain how they validated the findings.

Additional comments

The work is an innovative step to develop a CNN-based hybrid model to study CT scan images. The proposed model could be used to analyze CT scan images in other fields of the health sciences. However, the paper lacks depth in the discussion and conclusion.

---

## Round 0.4 · Major Revisions

Dear Authors,

Thank you for responding to the reviewers' comments. However, you have ignored my comments (below). Please revise the paper according to these comments and resubmit it again.

Editor Comments:

Please consider these comments and pay more attention to:
- highlighting the paper contributions of this paper. You may need to explain the hybrid nature of the proposed model.
- add more up-to-date references and discuss the research problems well in the introduction section.
- improve the quality of all figures used.
- Proofread the paper to improve the English writing. Editing the paper with the help of a professional proofreader will be useful.

Kind regards,
Faisal Saeed
Academic Editor

---

## Round 0.5 · accepted · Accept

Thank you for addressing the reviewers' and editor's comments. The paper is acceptable now.